# Bezafibrate Reduces Elevated Hepatic Fumarate in Insulin-Deficient Mice

**DOI:** 10.3390/biomedicines10030616

**Published:** 2022-03-06

**Authors:** Andras Franko, Martin Irmler, Cornelia Prehn, Silke S. Heinzmann, Philippe Schmitt-Kopplin, Jerzy Adamski, Johannes Beckers, Jürgen-Christoph von Kleist-Retzow, Rudolf Wiesner, Hans-Ulrich Häring, Martin Heni, Andreas L. Birkenfeld, Martin Hrabě de Angelis

**Affiliations:** 1Division of Diabetology, Endocrinology and Nephrology, Department of Internal Medicine IV, University Hospital Tübingen, 72076 Tuebingen, Germany; andras.franko@med.uni-tuebingen.de (A.F.); hu.haering@gmail.com (H.-U.H.); martin.heni@med.uni-tuebingen.de (M.H.); andreas.birkenfeld@med.uni-tuebingen.de (A.L.B.); 2Institute for Diabetes Research and Metabolic Diseases of the Helmholtz Centre Munich, University of Tübingen, 72076 Tuebingen, Germany; 3German Center for Diabetes Research (DZD e.V.), 85764 Neuherberg, Germany; beckers@helmholtz-muenchen.de; 4Institute of Experimental Genetics, Helmholtz Zentrum München, Deutsches Forschungszentrum für Gesundheit und Umwelt (GmbH), 85764 Neuherberg, Germany; martin.irmler@helmholtz-muenchen.de (M.I.); info.adamski@gmx.org (J.A.); 5Metabolomics and Proteomics Core (MPC), Helmholtz Zentrum München, 85764 Neuherberg, Germany; prehn@helmholtz-muenchen.de; 6Research Unit Analytical BioGeoChemistry, Helmholtz Zentrum München, 85764 Neuherberg, Germany; silke.heinzmann@helmholtz-muenchen.de (S.S.H.); schmitt-kopplin@helmholtz-muenchen.de (P.S.-K.); 7Department of Biochemistry, Yong Loo Lin School of Medicine, National University of Singapore, Singapore 117597, Singapore; 8Institute of Biochemistry, Faculty of Medicine, University of Ljubljana, 1000 Ljubljana, Slovenia; 9Chair of Experimental Genetics, Technical University of Munich, 85354 Freising, Germany; 10Center for Physiology and Pathophysiology, Institute of Vegetative Physiology, University of Köln, 50931 Cologne, Germany; juergen-christoph.von-kleist-retzow@uk-koeln.de (J.-C.v.K.-R.); rudolf.wiesner@uk-koeln.de (R.W.); 11Department of Pediatrics, Faculty of Medicine and University Hospital Cologne, University of Cologne, 50931 Cologne, Germany; 12Cologne Excellence Cluster on Cellular Stress Responses in Aging-associated Diseases (CECAD), University of Köln, 50931 Cologne, Germany; 13Center for Molecular Medicine Cologne, University of Köln, 50931 Cologne, Germany; 14Chair of Experimental Genetics, Center of Life and Food Sciences, Weihenstephan, Technische Universität München, 85354 Freising, Germany

**Keywords:** bezafibrate, diabetes, fumarate, insulin resistance, lysophosphatidylcholine

## Abstract

Glucotoxic metabolites and pathways play a crucial role in diabetic complications, and new treatment options which improve glucotoxicity are highly warranted. In this study, we analyzed bezafibrate (BEZ) treated, streptozotocin (STZ) injected mice, which showed an improved glucose metabolism compared to untreated STZ animals. In order to identify key molecules and pathways which participate in the beneficial effects of BEZ, we studied plasma, skeletal muscle, white adipose tissue (WAT) and liver samples using non-targeted metabolomics (NMR spectroscopy), targeted metabolomics (mass spectrometry), microarrays and mitochondrial enzyme activity measurements, with a particular focus on the liver. The analysis of muscle and WAT demonstrated that STZ treatment elevated inflammatory pathways and reduced insulin signaling and lipid pathways, whereas BEZ decreased inflammatory pathways and increased insulin signaling and lipid pathways, which can partly explain the beneficial effects of BEZ on glucose metabolism. Furthermore, lysophosphatidylcholine levels were lower in the liver and skeletal muscle of STZ mice, which were reverted in BEZ-treated animals. BEZ also improved circulating and hepatic glucose levels as well as lipid profiles. In the liver, BEZ treatment reduced elevated fumarate levels in STZ mice, which was probably due to a decreased expression of urea cycle genes. Since fumarate has been shown to participate in glucotoxic pathways, our data suggests that BEZ treatment attenuates the urea cycle in the liver, decreases fumarate levels and, in turn, ameliorates glucotoxicity and reduces insulin resistance in STZ mice.

## 1. Introduction

In a previous study, we investigated the beneficial effect of bezafibrate (BEZ) in a rodent model of diabetes [1]. BEZ is a pan-peroxisome proliferator-activated receptor (PPAR) activator, and like many other fibrates, it is used to reduce high circulating lipid levels [2,3]. In order to evoke insulin-deficient diabetes in mice, the animals were injected with streptozotocin (STZ), which destroys the pancreatic beta cells [4]. In addition to the STZ-injected mice, a control (Con) group was also studied without STZ injections. Furthermore, half of the animals were treated with a BEZ-containing diet or received a standard diet (SD). By using these four different mouse groups, we analyzed (i) the effect of the insulin-deficient diabetic state by comparing the diabetic STZ, SD mice with the Con, SD animals and (ii) the effect of BEZ comparing STZ, BEZ mice with STZ, SD animals. This previous study showed that BEZ treatment markedly attenuated hyperglycemia, decreased plasma lipids as well as improved glucose and insulin tolerance; however, the underlying molecular mechanisms were not completely understood [1]. 

PPARα, PPARγ and PPARδ are key transcription factors regulating the gene expression of many targets, which are implicated in lipid and glucose metabolism [5]. In the context of diabetes, the activation of all three PPARs by pan-PPAR agonists is currently under intensive research [6]. In several clinical studies, PPARα activation with fibrates ameliorated anti-diabetic microvascular disorders [7]. Patients with type 2 diabetes treated with BEZ showed improved lipid profiles, however, some other parameters like albuminuria remained unchanged [8]. Although fibrate treatment does not alter all clinical endpoints [9], BEZ was shown to ameliorate several cardiovascular events in clinical studies [10]. Interestingly, in clinical studies, which investigated the consequences of BEZ treatment in patients with type 2 diabetes, BEZ treatment improved many parameters. Flory and colleagues found that patients treated with BEZ had a lower hazard for incident diabetes compared with patients who got other fibrates [11]. Furthermore, Jones and colleagues observed that in comparison with a placebo group, patients with type 2 diabetes who were treated with BEZ showed a better glucose tolerance and an improved serum lipid profile [12]. These studies suggest that BEZ has potential anti-diabetic effects, but the underlying molecular pathways remain unclear. 

In order to identify the molecular pathways which are responsible for the beneficial effects of BEZ in diabetes, the current study was set up. In this study, we investigated the gene expression and metabolite levels of emerging pathways, which may possibly help to explain the improved glucose metabolism upon BEZ application. Clinical and preclinical investigations reported that BEZ has various tissue-specific effects affecting the liver, skeletal muscle and white adipose tissue (WAT) [13,14,15]. Therefore, we performed a systematic analysis of plasma, liver, skeletal muscle and WAT samples of BEZ-treated animals. Since glucose metabolism is tightly interconnected with inflammatory and insulin signaling pathways as well as lipid and hormone metabolism [16], we studied the gene expression of these major metabolic pathways. Furthermore, comprehensive metabolite profiling was performed using targeted and non-targeted metabolomics approaches, which involved a high number of lipid species and metabolites. Among these metabolites, we finally focused on fumarate, since a high fumarate level was shown to cause glucotoxic effects in skeletal muscle and adipose tissues; however, the biological relevance of fumarate in the liver remains elusive [17,18]. Since mitochondria have a central metabolic role in insulin resistance and diabetes [19], the activities of mitochondrial enzymes were separately measured in the liver. 

## 2. Materials and Methods

### 2.1. Animal Studies

Animal studies are described in our previous article [1]. Briefly, male C57BL/6N mice received a standard diet. Twelve-week-old mice were injected with 60 mg/kg STZ, whereas control mice were not treated. Two weeks later, the BEZ group received a standard diet supplemented with 0.5% (*w*/*w*) BEZ (B7273; Sigma-Aldrich, St. Louis, MI, USA) for eight weeks, while another group remained on a standard diet (SD group). Animals were sacrificed at the age of 22 weeks. Plasma triglycerides, nonesterified fatty acids, insulin level, blood glucose, quantitative insulin sensitivity check index (QUICKI), intraperitoneal glucose and insulin tolerance tests were measured as described previously [1]. 

### 2.2. Transcriptome Analysis

The transcriptome analysis was done as described previously [1]. Briefly, total RNA was isolated, amplified and hybridized on Affymetrix Mouse Gene 1.0 ST arrays. The statistical analysis and the pathway analysis were performed as shown previously [1]. Array data were uploaded as GSE179719, GSE39752 and GSE79008 databases.

### 2.3. Targeted Metabolomics with the AbsoluteIDQ p180 Kit

Flow injection-electrospray ionization-tandem mass spectrometry (FIA-ESI-MS/MS) and liquid chromatography-electrospray ionization-tandem mass spectrometry (LC-ESI-MS/MS) were performed, and the Absolute*IDQ*^TM^ p180 Kit (BIOCRATES Life Sciences AG, Innsbruck, Austria) was applied as published previously [20]. In brief, either 10 µL of plasma or 10 µL of the freshly prepared tissue homogenate was applied to the assay. Tissue homogenate was prepared as follows: Frozen WAT, liver or skeletal muscle samples were weighted into homogenization tubes with ceramic beads (1.4 mm). For metabolite extraction, to 1 mg of frozen liver tissue, 6 µL of a dry ice-cooled mixture of ethanol/phosphate buffer were added (85/15 *v*/*v*), instead of 3 µL to 1 mg of frozen WAT or muscle tissue, respectively. Tissue samples were homogenized and supernatants were applied for metabolite measurements. Tissue metabolite concentrations are given in pmol/mg wet weight, whereas concentrations in plasma are in μM. Statistics were done with MeV using a Wilcoxon–Mann–Whitney test with a false discovery rate of 10%.

### 2.4. Nuclear Magnetic Resonance (NMR)-Based Non-Targeted Metabolomics

The non-targeted metabolomics approach of both liver and plasma samples was undertaken using NMR spectroscopy. For the aqueous metabolites of the liver, we used 50 mg liver tissue and 1 mL extracted H_2_O as the extraction solvent. Extraction was achieved by the use of ceramic beads (NucleoSpin, Macherey–Nagel, Dueren, Germany) and followed by a homogenization (3 min at 30 1/s in a TissueLyser II (Qiagen, Hilden, Germany)) and centrifugation (5 min, 13,000× *g*). An aliquot of 120 µL of the supernatant was mixed with 60 μL NMR buffer (90% D_2_O, 500 mM PO_4_ buffer with 0.1% trimethylsilyl-tetradeuteropropionic acid (TSP), pH 7.4). Plasma sample preparation consisted of mixing 60 µL of plasma with 120 µL of saline D_2_O buffer (0.9% NaCl). NMR analysis happened immediately and in a randomized order. 

NMR spectra were measured with a Bruker 800 MHz spectrometer operating at 800.35 MHz. A standard one-dimensional pulse sequence (*noesypr1d*) delivered an overview of all molecules. The acquisition parameters were as follows: water suppression irradiation during recycle delay (2 s), mixing time = 200 ms, 90° pulse = 9 μs. We collected 512 scans into 64 K data points with a spectral width of 12 ppm. Plasma samples were analyzed with a Carr–Purcell–Meiboom–Gill pulse sequence (*cpmgpr1d*) to obtain an overview of small molecules. For metabolite identification, we analyzed a representative sample of both liver extract and plasma with a series of 2-dimensional NMR analyses, as further specified in [21]. The software TopSpin 3.2 (Bruker BioSpin, Ettlingen, Germany) was used for processing, i.e., Fourier transformation, manual phasing, baseline correction and calibration to TSP (δ 0.00) for liver or β-glucose (δ 4.64), for plasma. Data were imported into Matlab software R2011b (Mathworks, Natick, MA, USA) and further processed, i.e., the water region removed and spectra normalized [22]. Principal component analysis and orthogonal projection on latent structures’ discriminant analysis were carried out to find metabolites that discriminate the disease and treatment groups. Both unsupervised and supervised analyses were carried out in Matlab and the visualization of relevant metabolites was done according to the method described in [23]. Selected metabolites were relatively quantified by the area under the curve integration. 

### 2.5. Mitochondrial Enzyme Activities

The liver tissue was homogenized in a homogenization buffer. Activities of mitochondrial respiratory chain complexes II, III and IV, as well as the tricarboxylic acid (TCA) enzymes citrate synthase, isocitrate dehydrogenase and fumarase were determined as described previously [24,25,26,27,28]. Briefly, all mitochondrial enzyme activities were measured spectrophotometrically using a Varian Cary 50 scan photometer under Vmax conditions at pH optima. For example, the activity of complex II by measuring the rate of dichlorophenol indophenol reduction triggered by succinate at 600–750 nm, the activity of complex III by measuring the activity of the antimycin-sensitive decyl ubiquinol (DUQH2) cytochrome c reductase at 550–540 nm, and the activity of complex IV by following the oxidation of reduced cytochrome c in the presence of a detergent (lauryl-maltoside) at 550–540 nm. 

## 3. Results

### 3.1. BEZ Reverted the Diabetes-Modified Gene Profile

We have previously found that BEZ treatment markedly attenuated hyperglycemia, decreased plasma lipids, and improved glucose and insulin tolerance tests in STZ-injected diabetic mice (Table 1 and [1]). 

In order to identify the key genes and their upstream regulators, which transmit the beneficial effects of BEZ, we analyzed the liver, white adipose tissue (WAT) and skeletal muscle samples of BEZ-treated mice using microarrays. In the liver samples, we identified 1771 genes whose expression was significantly altered in the STZ, SD vs. Con, SD comparison, whereas, in the STZ, BEZ vs. STZ, SD comparison the expression of 3603 genes were significantly altered (Figure 1A). In the WAT samples, there were 3448 significantly regulated genes between the STZ, SD and Con, SD groups, whereas 476 genes were significantly altered between the STZ, BEZ and STZ, SD groups (Figure 1A). In the skeletal muscle samples, we observed 4701 significantly different expressed genes in the STZ, SD vs. Con, SD comparison, whereas only 34 genes were differently expressed between the STZ, BEZ and STZ, SD groups (Figure 1A). These genes were further analyzed by the Ingenuity pathway analysis (IPA) tool with the focus of the inverse pathway (for e.g., pathways, which are upregulated in the STZ, SD vs. Con, SD comparison but are downregulated in the STZ, BEZ vs. STZ, SD comparison and vice versa) (Figure 1A). The IPA upstream regulator analysis is used to identify the cascade of upstream transcriptional regulators that can explain the observed gene expression changes in a given dataset. 

**Figure 1 biomedicines-10-00616-f001:**
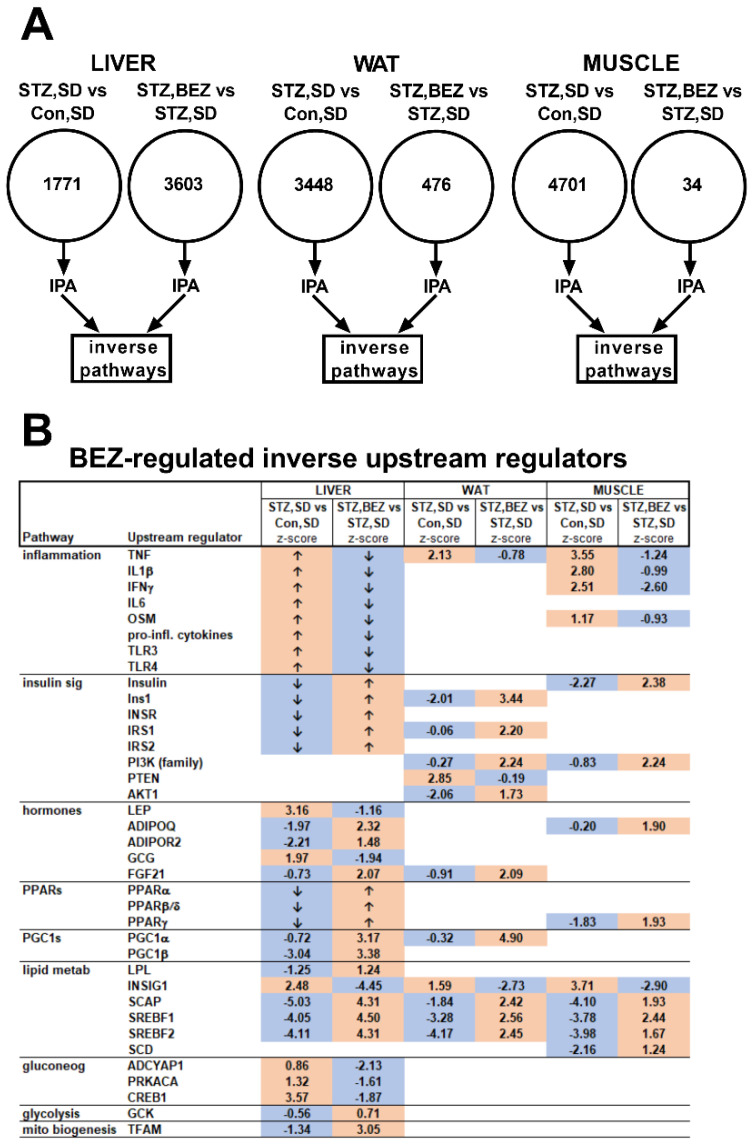
Microarray analysis in liver, white adipose tissue (WAT) and skeletal muscle samples of BEZ-treated STZ mice. (**A**) Schematic representation of microarray results. The numbers given in the circles depict the number of significantly differently expressed genes comparing STZ, SD vs. Con, SD as well as STZ, BEZ vs. STZ, SD, respectively. The statistical analysis was performed using significance analysis of microarrays algorithm with FDR of 10%. Upstream regulator tool of ingenuity pathway analysis (IPA) was applied on these significant genes with the focus on inverse pathways, *n* = 5–7. (**B**) Inverse upstream regulator pathways of IPA analysis between STZ, SD vs. Con, SD as well as STZ, BEZ vs. STZ, SD comparisons. Numbers and arrows denote z-scores of upstream regulator IPA pathways. The z-scores > 0 represent activated, whereas z-scores < 0 represent inhibited pathways. Values > 2 or <−2 indicate significant results, whereas values between −2 and 2 depict trends. “↑” or “↓” arrows show activated or inhibited upstream regulators described in [1] and the publisher granted permission to reuse the data. Orange or blue colors indicate activated or inhibited pathways, respectively. The theoretical order of the identified pathways, which are possibly involved in insulin sensitivity, glucose and lipid metabolism, are shown in Figure 2. Abbreviations: TNF: tumor necrosis factor, IL: interleukin, IFN: interferon, OSM: oncostatin M, pro-inf: pro-inflammatory, TLR: toll-like receptor, Ins1: insulin 1, INSR: insulin receptor, IRS: insulin receptor substrate, PI3K: phosphatidylinositol-4,5-bisphosphate 3-kinase, PTEN: phosphatase and tensin homolog, AKT: AKT serine/threonine kinase, LEP: leptin, ADIPOQ: adiponectin, ADIPOR: adiponectin receptor, GCG: glucagon, FGF: fibroblast growth factor, PPAR: peroxisome proliferator-activated receptor, PGC: peroxisome proliferator-activated receptor gamma coactivator, LPL: lipoprotein lipase, INSIG: insulin-induced gene, SCAP: SREBF chaperone, SREBF: sterol regulatory element-binding transcription factor, SCD: stearoyl-CoA desaturase, ADCYAP1: adenylate cyclase activating polypeptide 1, PRKACA: protein kinase A, CREB: cAMP response element-binding protein, GCK: glucokinase, TFAM: mitochondrial transcription factor A.

The upstream regulator IPA analysis showed that the inflammatory upstream regulators TNF, IL1β and IFNγ were activated in the diabetic state, but got reverted upon BEZ treatment (Figure 1B). Upstream regulators, which are involved in insulin signaling (for e.g., insulin, IRS1, PI3K) were inhibited in the diabetic animals but were activated in the BEZ group (Figure 1B). The diabetic state was characterized by inhibited hormonal (adiponectin and FGF21), transcriptional factor (PPARs) and transcriptional factor coactivator (PGC1α/β) upstream regulators, and BEZ activated these upstream regulators (Figure 1B). Furthermore, BEZ reverted the upstream regulators, which are participating in lipid metabolism (Figure 1B). In the liver, BEZ inhibited gluconeogenic upstream regulators, which became activated in the diabetic state. Furthermore, BEZ activated TFAM, which is considered to be a key factor during mitochondrial biogenesis (Figure 1B). These results suggest that BEZ attenuates the gene expression of inflammatory and gluconeogenic genes and induces the expression of genes regulating insulin signaling, PPARs, PGCs and lipid metabolism. The strongest change in the upstream regulators was observed in the liver, however, the data indicate that WAT and skeletal muscle are also important BEZ target tissues. 

### 3.2. BEZ Increased Levels of Lysophosphatidylcholines (lysoPC)

In order to determine pivotal lipid metabolites, which may possibly regulate the effect of BEZ, a targeted metabolomics approach was applied. This analysis also included the class of lysoPCs, which are linked to glucose and insulin metabolism [29,30]. Between the STZ, SD vs. Con, SD and STZ, BEZ vs. STZ, SD comparisons, 54, 72 and 37 common metabolites which significantly changed were identified in the plasma, liver and skeletal muscle samples, respectively (ref. [1], Appendix A, Figure 3A). Among these, 52, 66 and 37 metabolites showed “inverse” regulation (e.g., increased level in the STZ, SD vs. Con, SD comparison and decreased level in the STZ, BEZ vs. STZ, SD comparison or vice versa, i.e., a reverting effect of BEZ on changes due to STZ) in the plasma, liver and skeletal muscle samples, respectively (Figure 3A). Since we were interested in common significantly changed inverse metabolites, which are similarly altered in plasma, liver and skeletal muscle, these common metabolites were further analyzed (Figure 3B,C). In the WAT, our analysis did not identify any significant metabolite changes (Appendix A). 

The analysis of common significantly changed metabolites showed that the diabetic state was associated with a reduced level of lysoPCs in the plasma, liver and skeletal muscle samples, and BEZ reverted these changes (Figure 3C), suggesting that BEZ regulates lysoPC levels. Since the alterations of the lysoPC profile in skeletal muscle and liver reflected the changes in plasma, these results indicate that skeletal muscle and liver are major sources for plasma lysoPCs. 

### 3.3. BEZ Improved the Circulating Lipid Profile and Reduced Hepatic Fumarate

Compared to targeted metabolomics, non-targeted metabolomics measures metabolites without prior selection and possibly captures additional metabolites. In order to further investigate possible metabolites, which could explain the effect of BEZ, we used a non-targeted metabolomics approach. BEZ reduced glucose levels in plasma and in the liver (Figure 4A,F), decreased plasma VLDL + LDL lipids, increased HDL lipid levels in STZ, BEZ vs. STZ, SD comparison (Figure 4B,C) and reduced overall lipid levels in the liver (Figure 4G). In STZ, BEZ animals’ creatine levels were diminished in plasma and liver (Figure 4D,H). Elevated hepatic glycogen, lactate and fumarate levels in STZ, SD mice were reduced in STZ, BEZ animals (Figure 4I–K). Furthermore, BEZ decreased pyruvate levels in plasma and increased hepatic succinate levels in STZ, BEZ animals (Figure 4E,L). These data demonstrate that the major metabolites, which are regulated by BEZ, are lipids and mitochondrial metabolites. 

### 3.4. BEZ Altered Hepatic Mitochondrial Enzyme Activities and Reduced Fumarase Activity 

Since some metabolites originating from mitochondrial activity were significantly changed in the liver of BEZ-treated animals, we focused on hepatic mitochondrial enzyme activities. The activity of citrate synthase was elevated in BEZ-treated animals while the activity of isocitrate dehydrogenase remained unchanged (Figure 5A,B). BEZ treatment modified the activities of mitochondrial Complex II, III and IV (Figure 5C–E). In STZ, SD mice, fumarase activity was increased, and was reduced in the STZ, BEZ animals (Figure 5F). These results show that BEZ alters the activities of mitochondrial enzymes. 

### 3.5. BEZ Reduced the Hepatic Transcripts of Urea Cycle Genes 

Various biochemical pathways are involved in the production of intracellular fumarate levels. In addition to the mitochondrial TCA cycle, the urea cycle is also a possible source. Therefore, we analyzed the hepatic transcript levels of urea cycle enzymes. Among the six analyzed genes, the transcript level of Cps1, Ass1, Asl and Arg1 were significantly elevated in STZ, SD mice compared to Con, SD animals (Figure 6A–F), while transcripts of all six urea cycle genes were significantly reduced in the STZ, BEZ mice (Figure 6A–F). The schematic summary of the urea cycle with the observed transcript changes is depicted in Figure 6G.

In summary, BEZ altered the gene expression profile in the liver, skeletal muscle and WAT, suggesting attenuated inflammatory pathways, enhanced insulin signaling, as well as induced lipid oxidation and synthesis (Figure 2).

Furthermore, BEZ increased the levels of lysoPCs in plasma, liver and skeletal muscle, as well as decreased the hepatic level of fumarate, probably via reduced transcription of genes encoding urea cycle enzymes. Higher levels of lysoPCs and a lower level of fumarate are possibly involved in the beneficial effects of BEZ in ameliorating glucotoxicity and reducing insulin resistance (Figure 7).

## 4. Discussion

We have previously found that BEZ, a pan-PPAR activator, remarkably reduced high blood glucose levels and improved plasma lipid profiles and insulin sensitivity in type 1 and type 2 diabetic mouse models [1,31]. In the current study, we performed a systematic analysis of gene expression and metabolite levels in various organs using insulin-deficient diabetic STZ mice in order to investigate the underlying molecular pathways, which are possibly responsible for the beneficial effects of BEZ in glucose metabolism. 

In summary, our current study revealed that in the diabetic STZ, SD mice the upstream regulators of inflammatory genes were activated and BEZ attenuated the changes in gene expression of these inflammatory regulators in STZ, BEZ animals. The upstream regulators of insulin signaling and lipid metabolism genes were inhibited in STZ, SD mice and BEZ activated these regulators in STZ, BEZ animals. Among the studied metabolites, the level of lysoPC was significantly decreased in STZ, SD mice and BEZ normalized the levels of these lipid species. In the liver, the diabetic state was characterized by increased glucose and fumarate levels, both of which were reverted after BEZ treatment. The activity of hepatic fumarase, which metabolizes fumarate, was elevated in STZ, SD animals and BEZ reverted fumarase activity. The expression of urea cycle enzymes was increased in the diabetic state, which was normalized in the BEZ-treated animals. 

Although the STZ injection has been used to model insulin-deficient type 1 diabetes since the sixties, it does not completely mirror the human type 1 diabetic situation [32]. The inflammatory components of type 1 diabetic patients, such as insulitis, are not necessarily triggered in the rodent STZ model used in our study. Although our approach, the multiple low dose STZ injection, was suggested to induce insulitis in rodents [4], we did not confirm the inflammatory components in the pancreatic beta cells. Therefore, the STZ model used in the C57BL/6 background rather resembles an insulin-deficient diabetic state, which may not represent the human type 1 diabetic situation with insulitis. Nevertheless, our previous study demonstrated a significantly lower beta-cell number in the STZ mice compared to control animals, suggesting that these mice well resemble human type 1 diabetes in terms of abolished beta-cell characteristics [1].

### 4.1. The BEZ-Modified Transcriptional Profile

The reduced regulators of inflammatory genes observed in the tissues of BEZ-treated animals are in line with previous findings, which showed that BEZ attenuates inflammatory processes [33,34]. FGF21 and adiponectin are described as insulin sensitizers, and a high leptin level is associated with insulin resistance [35]. BEZ activated FGF21 and adiponectin upstream regulators, as well as inhibited leptin. These results are consistent with previous studies, which reported elevated circulating FGF21 and adiponectin as well as reduced leptin levels in BEZ-treated subjects and animals [36,37,38]. The expression pattern of FGF21, adiponectin and leptin in the BEZ-treated mice indicates an improved hepatic insulin sensitivity [35]. Lipid metabolism is regulated tightly at the transcriptional level. PPARs are major activators of fatty acid/lipid oxidation, and SREBPF is a key regulator of lipid synthesis [39,40]. Since in the BEZ-treated animals the PPARs and SREBFs upstream regulators were higher, these data suggest that BEZ enhanced both lipid oxidation (via PPARs) and lipid synthesis (via SREBFs), therefore the presence of futile lipid cycling and thus energy wasting is conceivable (Figure 2). The increased activation of SREBFs could be an indirect effect of BEZ and might be a consequence of improved insulin sensitivity [41]. The microarray data also showed that in addition to the liver, WAT and skeletal muscle are also important BEZ target tissues. In summary, these results indicate that BEZ attenuated inflammatory pathways, enhanced insulin signaling, as well as induced lipid oxidation but also synthesis in the liver, WAT and skeletal muscle (Figure 2 and Figure 7).

### 4.2. The BEZ-Modified Lipid Profile

In the diabetic STZ mice, BEZ decreased plasma VLDL + LDL lipids and increased HDL-lipids, as expected from previous studies [33,42]. Since BEZ was already described to alter these lipids, in this study, we focused on a less well-characterized lipid class. The lysoPCs are major plasma lipids, which are important cell-signaling molecules in immune, hepatic, adipose, muscle and pancreatic beta cells [43]. In diabetic STZ mice, we observed a significant reduction in many lysoPC species in plasma, skeletal muscle and liver—consistent with recent studies—which detected low circulating lysoPC levels in samples of type 1 diabetic progressor subjects compared with non-progressors [44], but also in pre-type 1 diabetic mice [45]. BEZ increased again the level of many lysoPCs in all investigated samples, in line with a previous report, which found elevated levels of lysoPC16:0 in plasma, skeletal muscle and liver of BEZ-treated mice [46]. 

The functions of lysoPCs have not been completely understood, and they depend on the target cell types and the molecular composition of lysoPC [47]. This could partly explain why lysoPCs are implicated in pro- as well as in anti-inflammatory processes [47]. It was demonstrated that in human skeletal muscle cells, lysoPCs reduced inflammation and ER stress [48]. This result suggests that the higher skeletal muscle lysoPC levels in the BEZ-treated animals could be involved in the attenuated inflammatory pathways (Figure 7). Furthermore, special lysoPCs could directly impact glucose metabolism, since the application of lysoPC16:0 activated adipocyte glucose uptake as well as lowered the blood glucose levels in murine models of diabetes [30,46]. Therefore, it is conceivable that the BEZ-induced elevation of tissue lysoPCs contributed to the increased plasma levels, which in turn improved glucose uptake (Figure 7) and reduced whole-body insulin resistance (Table 1 and [1]). 

### 4.3. The BEZ-Modified Metabolite Profile

In the liver of STZ mice, BEZ decreased the high hepatic glucose level. In the long term, increased intracellular glucose levels trigger chronic glucotoxicity/hyperglycemic stress, which alters intracellular glucose and lipid pathways and promotes cellular dysfunction and eventually death [49,50]. Hyperglycemic stress is associated with many harmful metabolic changes, including enhanced reductive stress, protein kinase C activation, polyol and hexosamine synthesis, enediol formation and advanced glycation end products pathways, which are all involved in the development of diabetic complications [51,52]. The glyceraldehyde-3-phosphate dehydrogenase (GAPDH) is a pivotal regulator of glycolysis, and upon inactivation of this enzyme, the level of glycolytic intermediates could accumulate, which in turn leads to enhanced polyols, advanced glycation end products, hexosamine and the protein kinase C pathways [17]. In diabetic rats, a decreased activity of GAPDH was found, and elevated fumarate levels and enhanced succination were identified as underlying mechanisms for the inactivation of GAPDH [17,53]. In the diabetic STZ mice, we observed a high hepatic fumarate level, which was reverted in the BEZ-treated animals. However, in the liver of STZ, SD mice, the activity of fumarase, which is responsible for the metabolization of fumarate, was increased, in accordance with our previous study [25].

We found that BEZ reduced hepatic fumarase activity. Moreover, the expression of genes encoding enzymes involved in the urea cycle, which is known to produce fumarate in the liver, was high in diabetic STZ mice and was normalized in BEZ-treated animals. These data indicate that BEZ reduces urea cycle enzyme activities, and thus decreases hepatic fumarate levels, which ameliorates glucotoxicity and in turn reduces whole-body insulin resistance (Table 1 and Figure 7). 

Elevated fumarate levels have already been associated with the progression of diabetes. In a rodent study, short-term STZ treatment resulted in an increased hepatic fumarate level, however, this was not evident after longer STZ applications [54]. Indeed, fumarase deletion in mouse pancreatic beta cells led to elevated fumarate levels and enhanced protein succination, which were associated with decreased insulin secretion and the development of diabetes [55]. In a diabetic mouse model, a high fumarate level was implicated in ER stress and glomerular dysfunction of the kidney, and intervention with an NADPH oxidase isoform 1/4 inhibitor decreased fumarate and improved kidney function in the treated animals [56,57]. 

Many of the BEZ-triggered molecular changes are probably regulated via PPARα, and indeed PPARα alters gene expression of urea cycle enzymes and thus modifies fumarate levels. PPARα knock-out mice were characterized by an enhanced gene expression of urea cycle enzymes [58] and increased hepatic fumarate content [59]. Furthermore, transactivation of PPARα resulted in a metabolic shift that favored lipid catabolism over proteins, and thus suppressed urea cycle gene expression [60]. 

## 5. Conclusions

Altogether, our data indicate that the BEZ-induced tissue-specific gene expression, lipid and metabolite profiles contributed to the ameliorated glucotoxicity, which possibly reduced insulin resistance in BEZ-treated diabetic mice (Figure 7).

## 6. Future Prospects

Since the applied STZ model does not completely resemble human type 1 diabetes, the beneficial effects of BEZ could be further verified in other rodent models like in non-obese diabetic (NOD) mice, which better mirrors the pancreatic inflammatory components of type 1 diabetes. 

## Figures and Tables

**Figure 2 biomedicines-10-00616-f002:**
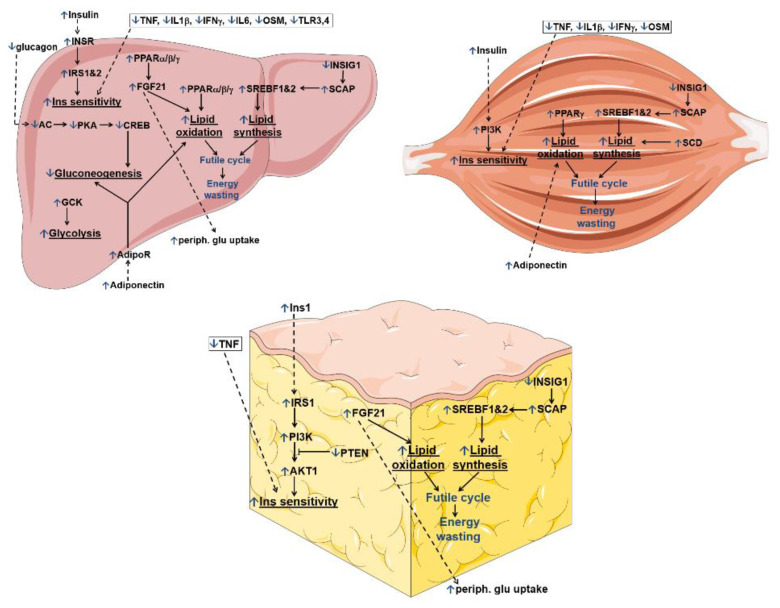
Theoretical order of the IPA transcriptional upstream regulator pathways, which are altered upon BEZ treatment. All IPA upstream regulator pathways of the liver, WAT and skeletal muscle tissues are shown in Figure 1. This figure summarizes those pathways, which are possibly involved in insulin (Ins) sensitivity, glucose as well as lipid metabolism. Blue “↑” or “↓” arrows show BEZ-activated or BEZ-inhibited pathways. Black arrows represent direct, whereas dashed arrows denote indirect regulations.

**Figure 3 biomedicines-10-00616-f003:**
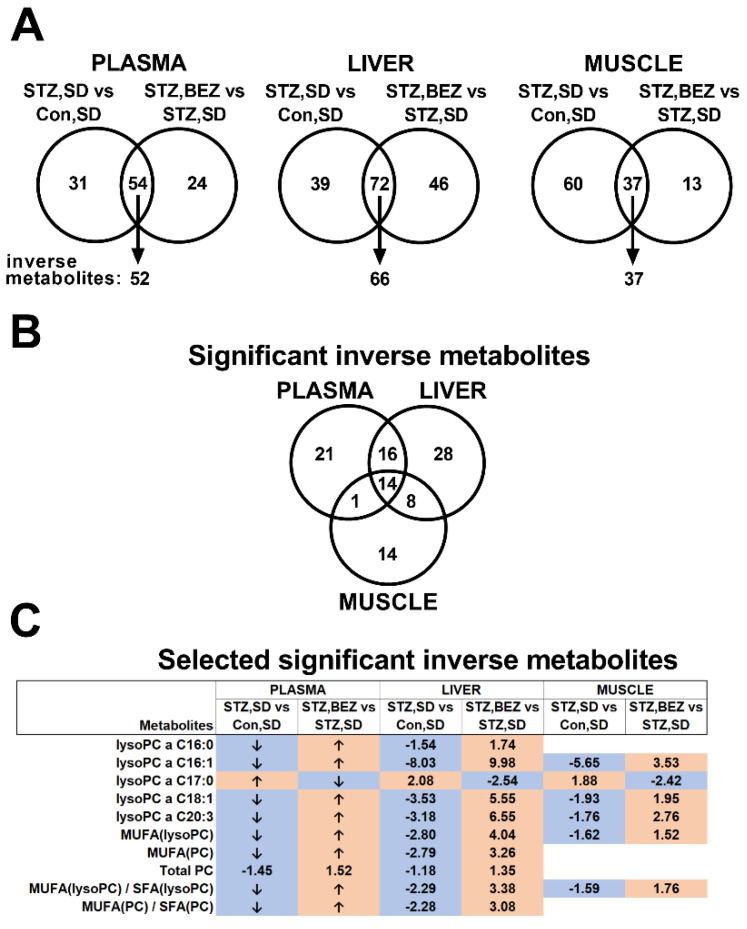
Targeted metabolomics analysis in plasma, liver and skeletal muscle samples of BEZ-treated STZ mice. (**A**) Schematic representation of targeted metabolomics results. Significantly altered common metabolites and metabolite ratios comparing STZ, SD vs. Con, SD and STZ, BEZ vs. STZ, SD, respectively, were identified in plasma, liver and skeletal muscle samples. Wilcoxon–Mann–Whitney test with FDR of 10% was applied, *n* = 5–7. Those metabolites and metabolite ratios, which showed inverse regulation, were further analyzed. (**B**) The number of significant inversely regulated metabolites in plasma, liver and skeletal muscle tissues, which are common in the analyzed tissues. (**C**) Fold changes of selected significantly inversely regulated metabolites are shown, which are common in the analyzed tissues. Fold changes were calculated by dividing the appropriate mean of groups. “↑” or “↓” arrows show increased or decreased metabolites already described in [1] and the publisher granted permission to reuse the data. Orange or blue colors indicate increased or decreased metabolites, respectively. Abbreviations: PC: phosphatidylcholine, lysoPC: lysophosphatidylcholine, SFA: saturated fatty acid, MUFA: monounsaturated fatty acid, a: acyl.

**Figure 4 biomedicines-10-00616-f004:**
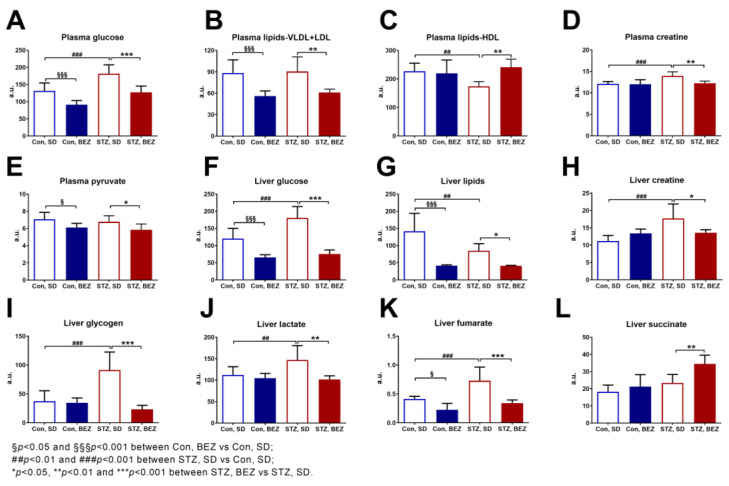
NMR-based metabolomics analyses in plasma and liver samples of BEZ-treated STZ mice. Metabolites measured in (**A**–**E**) plasma and (**F**–**L**) liver samples. Columns represent averages ± standard deviation; *n* = 6–8. ANOVA with post hoc Holm–Sidák multiple comparison test was applied. Abbreviations: VLDL: very-low-density lipoprotein, LDL: low-density lipoprotein, HDL: high-density lipoprotein, a.u.: arbitrary unit.

**Figure 5 biomedicines-10-00616-f005:**
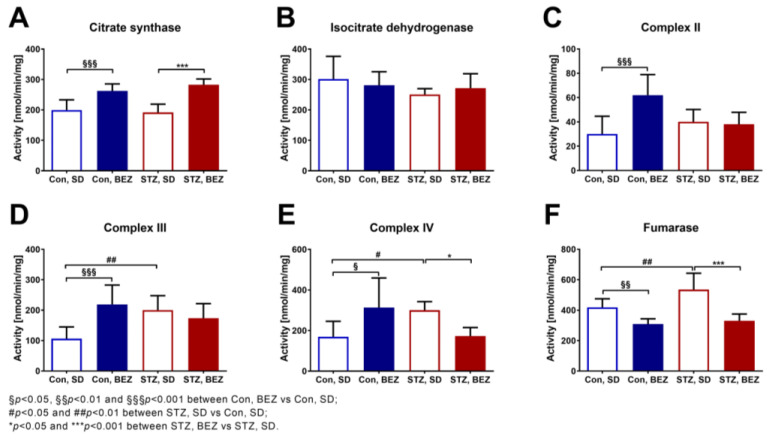
Mitochondrial enzyme activities were measured in liver samples of BEZ-treated STZ mice (**A**–**F**). Columns represent averages ± standard deviation; *n* = 6–8. ANOVA with post hoc Holm–Sidák multiple comparison test was applied.

**Figure 6 biomedicines-10-00616-f006:**
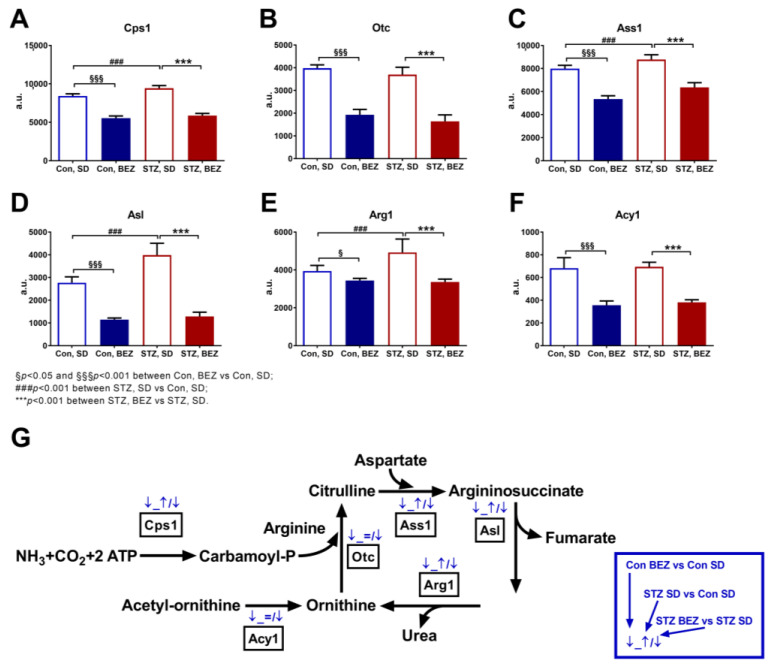
Transcript levels of urea cycle enzymes in liver samples of BEZ-treated STZ mice. (**A**–**F**) Columns represent averages ± standard deviation; *n* = 5–7. ANOVA with post hoc Holm–Sidák multiple comparison test was applied. (**G**) This figure summarizes the hepatic urea cycle genes, which were significantly differentially expressed upon BEZ treatment. The first blue arrow shows the comparison of Con, BEZ vs. Con, SD; the second blue arrow shows the comparison of STZ, SD vs. Con, SD; and the third blue arrow shows the comparison of STZ, BEZ vs. STZ, SD groups (for details see the blue rectangle on the right side). Blue “↑” or “↓” arrows show BEZ-activated or BEZ-inhibited genes. Abbreviations: Cps1: carbamoyl-phosphate synthase 1, Otc: ornithine transcarbamylase, Ass1: argininosuccinate synthase 1, Asl: argininosuccinate lyase, Arg1: arginase 1, Acy1: aminoacylase 1, a.u.: arbitrary unit.

**Figure 7 biomedicines-10-00616-f007:**
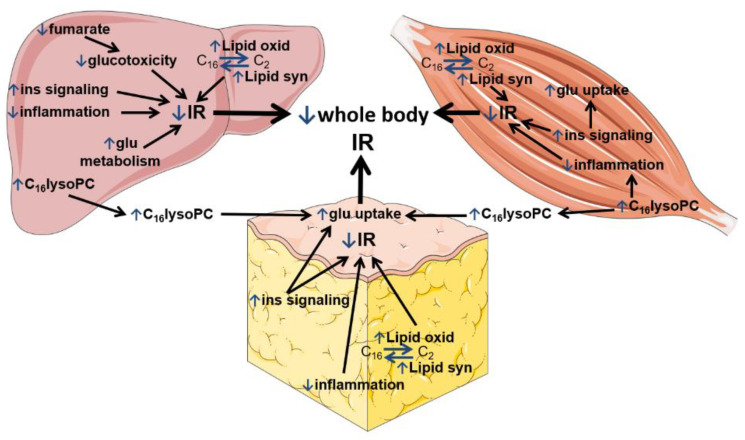
BEZ-modified metabolic pathways, which reduce whole-body insulin resistance (IR). The figure summarizes the beneficial effects of BEZ in the liver (left side), skeletal muscle (right side) and white adipose tissue (in the middle), which are possibly involved in insulin sensitivity, glucose and lipid metabolism reducing whole-body insulin resistance (IR). Blue “↑” or “↓” arrows show BEZ-activated or BEZ-inhibited pathways. Abbreviations: lipid oxid: lipid oxidation, lipid syn: lipid synthesis, ins: insulin, glu: glucose, lysoPC: lysophosphatidylcholine, IR: insulin resistance.

**Table 1 biomedicines-10-00616-t001:** Metabolic parameters of BEZ-treated STZ mice.

Parameters	Con, SD	Con, BEZ	STZ, SD	STZ, BEZ
**Body weight [g]**	35.36 ± 3.15	29.23 ± 1.11 ^§§^	22.27 ± 3.38 ^###^	25.16 ± 3.25
**Blood glucose [mg/dL]**	141.71 ± 11.10	127.86 ± 13.78	534.67 ± 58.02 ^###^	320.40 ± 78.83 ***
**Plasma insulin [µg/L]**	2.556 ± 0.944	0.563 ± 0.393 ^§§§^	0.264 ± 0.157 ^###^	0.261 ± 0.161
**QUICKI**	0.256 ± 0.010	0.275 ± 0.011 ^§§^	0.233 ± 0.011 ^##^	0.259 ± 0.014 ***
**GTT AUC [a.u.]**	22,501 ± 4667	14,638 ± 1334 ^§^	54,253 ± 8191 ^###^	38,429 ± 8543 ***
**ITT AUC [a.u.]**	4823 ± 674	2875 ± 434 ^§§§^	4748 ± 389	3078 ± 302 ***
**Plasma triglyceride [mg/dL]**	112.17 ± 38.63	40.11 ± 7.09 ^§^	149.08 ± 79.29	34.18 ± 3.84 ***
**Plasma NEFA [mmol/L]**	0.409 ± 0.087	0.427 ± 0.128	0.630 ± 0.226 ^#^	0.416 ± 0.035 *

Con: control mice, STZ: streptozotocin-injected insulin-deficient mice, SD: standard diet, BEZ: bezafibrate diet, QUICKI: quantitative insulin sensitivity check index, GTT: glucose tolerance test, ITT: insulin tolerance test, AUC: area under the curve, NEFA: nonesterified fatty acids. Numbers represent averages ± standard deviation; *n* = 5–8. ANOVA with post hoc Holm–Sidák multiple comparison test was used to calculate statistical significance, which was assumed as *p* < 0.05. ^§^ *p* < 0.05, ^§§^ *p* < 0.01 and ^§§§^ *p* < 0.001 between Con, BEZ vs. Con, SD; ^#^ *p* < 0.05, ^##^ *p* < 0.01 and ^###^ *p* < 0.001 between STZ, SD vs. Con, SD; * *p* < 0.05 and *** *p* < 0.001 between STZ, BEZ vs. STZ, SD. Data have been published in [1] and the publisher granted permission to reuse the data.

## Data Availability

Array data were submitted to the National Center for Biotechnology Information GEO (Gene Expression Omnibus) database (GSE39752, GSE79008, GSE179719).

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
