# Peer review of "Bezafibrate Reduces Elevated Hepatic Fumarate in Insulin-Deficient Mice"

_biomedicines, 2022, doi:10.3390/biomedicines10030616_

Round 1

Reviewer 1 Report

  • This manuscript is showing 32% similarity. Please reduce it to less than 20%.
  • The abstract must show some important results of the research. Please change the abstract.
  • This sentence is not a good start for an abstract. “In addition to high glucose levels, the diabetic state is also characterized by other altered metabolites, which are critical for glucotoxic pathways”. Delete the sentence, please.
  • The introduction is too short and must include some review of other studies and then what is the gap and why you are performing this research.
  • Section 2.4: what is the use of ceramic beads? Is it possible without using the ceramic beads to perform the experiment?
  • Section 2.5: Please briefly explain the methodology.
  • Compare the results with other studies.
  • Some references are too old and are better to be replaced with some newly published papers such as ref. 6, 9, 16,17,18, …
  • Please add “future prospects” at the end of the manuscript.
  • It is suggested to use the following reference in this study:

Sabbagh, F., Muhamad, I. I., Niazmand, R., Dikshit, P. K., & Kim, B. S. (2022). Recent progress in polymeric non-invasive insulin delivery. International Journal of Biological Macromolecules.

Author Response

  1. This manuscript is showing 32% similarity. Please reduce it to less than 20%.

The similarity was extensively reduced in the manuscript with the help of Ms Yang and we tried our best to delete all unnecessary information. The similarity is now around 20%. Most of the remaining similarities are found in the method and figure legend sections, which describe the experimental setup and the statistical analysis. These details are needed for the reader to understand how the experiments were performed and how the data were evaluated.

  1. The abstract must show some important results of the research. Please change the abstract.

The abstract was modified and the requested information are described in details as following:

“Glucotoxic metabolites and pathways play a crucial role in diabetic complications and new treatment options which improve glucotoxicity are highly warranted. In this study we analyzed bezafibrate (BEZ) treated, streptozotocin (STZ) injected mice, which showed improved glucose metabolism compared to untreated STZ animals. In order to identify key molecules and pathways which participate in the beneficial effects of BEZ, we studied plasma, skeletal muscle, white adipose tissue (WAT) and liver samples using non-targeted metabolomics (NMR spectroscopy), targeted metabolomics (mass spectrometry), microarrays and mitochondrial enzyme activity measurements with a particular focus on the liver. The analysis of muscle and WAT demonstrated that STZ treatment elevated inflammatory pathways and reduced insulin signaling and lipid pathways, whereas BEZ decreased inflammatory pathways and increased insulin signaling and lipid pathways, which can partly explain the beneficial effects of BEZ on glucose metabolism. Furthermore, lysophosphatidylcholine levels were lower in the liver and skeletal muscle of STZ mice, which were reverted in BEZ-treated animals. BEZ also improved circulating and hepatic glucose and lipid profiles. In the liver, BEZ treatment reduced elevated fumarate levels in STZ mice, which was probably due to decreased expression of urea cycle genes. Since fumarate has been shown to participate in glucotoxic pathways, our data suggest that BEZ treatment attenuates the urea cycle in the liver, decreases fumarate levels and in turn ameliorates glucotoxicity and reduces insulin resistance in STZ mice”

  1. This sentence is not a good start for an abstract. “In addition to high glucose levels, the diabetic state is also characterized by other altered metabolites, which are critical for glucotoxic pathways”. Delete the sentence, please.

The sentence was rephrased as:

“Glucotoxic metabolites and pathways play a crucial role in diabetic complications and new treatment options which improve glucotoxicity are highly warranted.”

  1. The introduction is too short and must include some review of other studies and then what is the gap and why you are performing this research.

We extended the Introduction and included novel articles and reviews as it was requested:

“PPARa, PPARg and PPARd are key transcription factors regulating gene expression of many targets, which are implicated in lipid and glucose metabolism [5]. In the context of diabetes, the activation of all three PPARs by pan-PPAR agonists are currently under intensive research [6]. In several clinical studies, PPARa activation with fibrates ameliorated antidiabetic microvascular disorders [7]. Patients with type 2 diabetes treated with BEZ showed improved lipid profiles, however some other parameters like albuminuria remained unchanged [8]. Although fibrate treatment does not alter all clinical endpoints [9], BEZ was shown to ameliorate several cardiovascular events in clinical studies [10]. Interestingly, in clinical studies, which investigated the consequences of BEZ treatment in patients with type 2 diabetes, BEZ treatment improved many parameters. Flory and col-leagues found that patients treated with BEZ had a lower hazard for incident diabetes compared with patients who got other fibrates [11]. Furthermore, Jones and colleagues observed that in comparison with a placebo group, patients with type 2 diabetes who were treated with BEZ showed better glucose tolerance and improved serum lipid profile [12]. These studies suggest that BEZ has potential anti-diabetic effects, but the underlying molecular pathways remain unclear.

In order to identify the molecular pathways, which are responsible for the beneficial effects of BEZ in diabetes, the current study was set up. In this study we investigated the gene expression and metabolite levels of emerging pathways, which may possibly help to explain the improved glucose metabolism upon BEZ application.”

  1. Section 2.4: what is the use of ceramic beads? Is it possible without using the ceramic beads to perform the experiment?

For cell disruption the application of ceramic beads in combination with homogenizers is a state-of-the-art technology, which is widely used (Dubacq, S. Nat Methods 13, i–iii, 2016). It is possible to avoid the beads when using other solvents (ice-cold methanol or water) where cell disruption and osmolysis might happen on a more chemical rather than physical level. However, we are not sure whether this method would be as complete as the method used by us. In this study, our particular focus was to analyze the hydrophilic fraction of liver metabolites and therefore we decided to use the previously described method with ceramic beads.

  1. Section 2.5: Please briefly explain the methodology.

The methods in section 2.5 are now described in more details:

“Briefly, all mitochondrial enzyme activities were measured spectrophotometrically using a Varian Cary 50 scan photometer under Vmax conditions at pH optima. For example: the activity of complex II by measuring the rate of dichlorophenol indophenol reduction triggered by succinate at 600-750 nm, the activity of complex III by measuring the activity of the antimycin-sensitive decyl ubiquinol (DUQH2) cytochrome c reductase at 550-540 nm and the activity of complex IV by following the oxidation of reduced cytochrome c in the presence of a detergent (lauryl-maltoside) at 550-540 nm.”

  1. Compare the results with other studies.

In the Discussion, we indeed compared the results of our study with previous findings. Briefly, in line 389-398 we compared our results of FGF21, adiponectin and leptin with the literature. In line 409-427 we compared our results of lysoPC with the literature and interpreted these data. In line 429-445 and 452-460 we compared our results of fumarate with the literature data with a particular focus on glucotoxic pathways and the development of diabetes. In line 461-466 we interpreted the data of the literature regarding the role of PPARalpha and its possible consequences.

  1. Some references are too old and are better to be replaced with some newly published papers such as ref. 6, 9, 16,17,18, …

In our scientific field, “older” references do not necessarily mean that they are outdated. Many “old” studies show outstanding clinical data. Before writing the manuscript we carefully assessed the literature and included all relevant articles in the manuscript. These previously included articles are unique and still considered to be relevant and they are very different in term of conclusion from the suggested references (6, 9, 16, 17, 18.). Therefore, it is not possible to exchange these previous references. Nevertheless, in the revised version of the manuscript we included many novel articles (see answer 4 of comment 4).

  1. Please add “future prospects” at the end of the manuscript.

The future prospects are included in the manuscript as following:

“Since the applied STZ model does not completely resemble the human type 1 diabetes, the beneficial effects of BEZ could be further verified in other rodent models like in non-obese diabetic (NOD) mice, which better mirrors the pancreatic inflammatory components of type 1 diabetes.”

  1. It is suggested to use the following reference in this study: Sabbagh, F., Muhamad, I. I., Niazmand, R., Dikshit, P. K., & Kim, B. S. (2022). Recent progress in polymeric non-invasive insulin delivery. International Journal of Biological Macromolecules.

As Reviewer 1 and 2 suggested, we extended the Introduction and included numerous new relevant references regarding the research topic (see answer 4 of comment 4.). Since our study describes the beneficial effect of Bezafibrate in a type 1 diabetic mouse model with the particular focus on key alterations in gene expression, lipid and metabolite levels, we think that this suggested publication about design of carriers for insulin delivery like non-invasive insulin delivery mechanisms including oral, transdermal, rectal, vaginal, ocular, and nasal does not fit in the scope of our manuscript.

Reviewer 2 Report

The paper “Bezafibrate reduces elevated hepatic fumarate in insulin-deficient mice" by Franko et al. is a rodent study with the aim of investigating changes in both gene expression and metabolite levels implied in glucose metabolism improvement, in rodent treated with bezafibrate.

The article is well written. The paper has a good design. The article is logically divided into sections and subsections. The references cited are relevant and adequate. The work has an average degree of novelty and of good interest to the readers.

Comment:

  • Introduction must be rewritten as a better background should be provided by the author. I do understand that this article is a continuum with the previous published, however, a better insight should be provided.

Author Response

The paper “Bezafibrate reduces elevated hepatic fumarate in insulin-deficient mice" by Franko et al. is a rodent study with the aim of investigating changes in both gene expression and metabolite levels implied in glucose metabolism improvement, in rodent treated with bezafibrate.

The article is well written. The paper has a good design. The article is logically divided into sections and subsections. The references cited are relevant and adequate. The work has an average degree of novelty and of good interest to the readers.

Comment:

Introduction must be rewritten as a better background should be provided by the author. I do understand that this article is a continuum with the previous published, however, a better insight should be provided.

We rewrote and extended the Introduction and included novel articles and reviews as it was requested:

“PPARa, PPARg and PPARd are key transcription factors regulating gene expression of many targets, which are implicated in lipid and glucose metabolism [5]. In the context of diabetes, the activation of all three PPARs by pan-PPAR agonists are currently under intensive research [6]. In several clinical studies, PPARa activation with fibrates ameliorated antidiabetic microvascular disorders [7]. Patients with type 2 diabetes treated with BEZ showed improved lipid profiles, however some other parameters like albuminuria remained unchanged [8]. Although fibrate treatment does not alter all clinical endpoints [9], BEZ was shown to ameliorate several cardiovascular events in clinical studies [10]. Interestingly, in clinical studies, which investigated the consequences of BEZ treatment in patients with type 2 diabetes, BEZ treatment improved many parameters. Flory and col-leagues found that patients treated with BEZ had a lower hazard for incident diabetes compared with patients who got other fibrates [11]. Furthermore, Jones and colleagues observed that in comparison with a placebo group, patients with type 2 diabetes who were treated with BEZ showed better glucose tolerance and improved serum lipid profile [12]. These studies suggest that BEZ has potential anti-diabetic effects, but the underlying molecular pathways remain unclear.

In order to identify the molecular pathways, which are responsible for the beneficial effects of BEZ in diabetes, the current study was set up. In this study we investigated the gene expression and metabolite levels of emerging pathways, which may possibly help to explain the improved glucose metabolism upon BEZ application.”

Reviewer 3 Report

Although study is well designed, I have a major concern about the strain used for study experimental T1D. C57Bl/6 strain don't develop any insulitis when used as a model for T1D using low doses of STZ for five consecutive days. (PMIDS: 154403 and 180605). Thus, concluding that BEZ altered hepatic fumarate in experimental T1D doesn't sound scientifically correct. 

Minor point: authors discussing about the insulin resistance in the manuscript, as per clinical T1D such phenomena in T1D patients occurred at a very late stage of the diseases, also, in a very few patients.

Author Response

Although study is well designed, I have a major concern about the strain used for study experimental T1D. C57Bl/6 strain don't develop any insulitis when used as a model for T1D using low doses of STZ for five consecutive days. (PMIDS: 154403 and 180605). Thus, concluding that BEZ altered hepatic fumarate in experimental T1D doesn't sound scientifically correct.

We completely agree with Reviewer 3 that STZ injection in the C57BL/6 mouse strain does not completely mirror the human type 1 diabetic state. We are aware that insulitis could be a missing process in the applied rodent model. Therefore, we never claimed that “BEZ altered hepatic fumarate in experimental T1D” as the reviewer stated. In the manuscript, we always used the terms “STZ mice” (see Abstract) or “insulin-deficient mice” or “insulin-deficient diabetic STZ mice” and never generalized our findings to “experimental T1D” as the reviewer stated. Nevertheless, we included now a short limitation paragraph regarding STZ injection in the Discussion and the possibility of missing inflammation in the beta-cells to make this point clear for the reader:

“Although the STZ injection has been used to model insulin-deficient type 1 diabetes since the sixties, it does not completely mirror the human type 1 diabetic situation [32]. The inflammatory components of type 1 diabetic patients like insulitis are not necessarily triggered in the rodent STZ model used in our study. Although our approach, the multiple low dose STZ injection, was suggested to induce insulitis in rodents [4], we did not confirm the inflammatory components in the pancreatic beta-cells. Therefore, the STZ model used in the C57BL/6 background rather resembles an insulin-deficient diabetic state, which may not represent the human type 1 diabetic situation with insulitis. Nevertheless, our previous study demonstrated a significantly lower beta-cell number in the STZ mice compared to control animals suggesting that these mice resemble well the human type 1 diabetes in term of abolished beta-cell characteristics [1].”

Furthermore, this was also pointed out in the Future prospects as the following:

“Since the applied STZ model does not completely resemble the human type 1 diabetes, the beneficial effects of BEZ could be further verified in other rodent models like in non-obese diabetic (NOD) mice, which better mirrors the pancreatic inflammatory components of type 1 diabetes.”

Minor point: authors discussing about the insulin resistance in the manuscript, as per clinical T1D such phenomena in T1D patients occurred at a very late stage of the diseases, also, in a very few patients.

We agree with the reviewer that insulin resistance in patients with T1D is a rare phenomenon and if it occurs then it is apparent in the late disease state. Nevertheless, the STZ model applied in our study resembles an overt diabetes, and the animals remained untreated for several weeks. In this background, insulin resistance is a common feature, which is not typical in patients, who get a quick treatment compensating diabetes. Furthermore, in a previous study of us (Franko A et al J Hepatol. 2014 Apr;60(4):816-23. PMID: 24291365), we indeed measured insulin resistance in STZ mice with the gold-standard euglycemic hyperinsulinemic clamp. These data showed that STZ treated mice expressed a markedly decreased whole body insulin sensitivity compared to untreated mice. This was accompanied by the reduced ability of insulin to suppress endogenous glucose production rates from basal values in STZ treated compared to untreated individuals. Therefore, we think, that insulin resistance is a key feature in the STZ model and the beneficial effect of BEZ is interesting to study.

Round 2

Reviewer 2 Report

The authors managed to improve the manuscript. It can be further processed for publication 

Round 3

Reviewer 2 Report

The author fulfilled all my request, I probably made a mistake by 
checking minor revision. I wished to write accept in present form. I 
apologize for the inconvenience.